# A Person-Centered Approach to Job Insecurity: Is There a Reciprocal Relationship between the Quantitative and Qualitative Dimensions of Job Insecurity?

**DOI:** 10.3390/ijerph20075280

**Published:** 2023-03-27

**Authors:** Sonia Nawrocka, Hans De Witte, Margherita Pasini, Margherita Brondino

**Affiliations:** 1Research Group Work, Organisational and Personnel Psychology, KU Leuven, 3000 Leuven, Belgium; 2Department of Human Sciences, University of Verona, 37129 Verona, Italy; 3Optentia Research Unit, North-West University, Vanderbijlpark 1900, South Africa

**Keywords:** quantitative job insecurity, qualitative job insecurity, within-person, random-intercept cross-lagged panel model, reciprocity

## Abstract

Radical transformations in the current work model induce qualitative job insecurity (i.e., a threat to job characteristics) and strengthen quantitative job insecurity (i.e., a threat to job loss). Both dimensions are separate yet interdependent work stressors. Although organisational changes are often the core source for both types of job insecurity, it is predominantly a subjective experience—individual perception ultimately determines the risk and the consequences of these threats. So far, the between-person analysis suggests that the relationship between the two dimensions is in both directions. However, it is not clear whether these associations also reflect within-person processes. This study proposes and tests the reciprocal relationship between quantitative and qualitative job insecurity at the within-person level. We employed a multiple indicator random-intercept cross-lagged panel model (RI-CLPM) to test these associations within-person while controlling for between-person differences. We used three-wave longitudinal data (6 months’ time lag) collected from a Belgian working population (N = 3694). The results suggest a unidirectional relationship (from quantitative to qualitative job insecurity). Furthermore, the results reveal significant within-person carry-over effects of quantitative job insecurity but not for qualitative job insecurity. Overall, these results suggest that a change in the experience of threats to job loss (i.e., higher-than-usual quantitative job insecurity) not only anticipates higher-than-usual threats to job loss (autoregressive paths) but also higher-than-usual threats to job characteristics (i.e., qualitative job insecurity), six months later. This study contributes to the ongoing discussion on how job insecurity dimensions influence each other. Given these results and the continuous changes to how we work, we call for further research to better understand the within-person processes of job insecurity development.

## 1. Introduction

In the progressively volatile labour market, increasing numbers of employees report worrying about the future of their work situation [1]. In addition, longitudinal research suggests that feelings of job insecurity persist over time, which indicates a continuous process in which the initial feelings of worry are sustained over time [2,3]. Accordingly, job insecurity is considered an omnipresent work stressor, with the European Union calling it a “main psychological hazard” [4]. This is not surprising since an overwhelming amount of evidence has linked job insecurity with adverse consequences for employees’ and organizations’ well-being [5,6]. Given the prevalence and severity of these consequences, it is urgent to understand how the continuous process of experiencing job insecurity is maintained.

The most prominent definition of job insecurity originates from Greenhalgh and Rosenblatt’s seminal article, in which they described job insecurity as an individual perception of a “potential loss of continuity in a job situation” [7]. That definition characterises job insecurity in broad terms that could be further classified as either a worry of losing current employment or a worry of losing the valued characteristics of that employment [8]. Therefore, in the recent literature, job insecurity has been predominantly defined as a two-dimensional construct, identifying quantitative and qualitative aspects of job insecurity, respectively. Although quantitative and qualitative job insecurity is two aspects of the same construct, they are theoretically and empirically distinct. To date, research that explicitly compared the quantitative and qualitative dimensions of job insecurity consistently found them to fit as separate dimensions (i.e., CFA model comparisons show that both constructs are empirically distinct) [9]. Furthermore, quantitative and qualitative job insecurity has a distinct relationship with theoretical predictors and outcomes of job insecurity [10,11,12,13]. That said they are considered separate yet closely related, psychological stressors [5].

In the current study, we examined the relationship between quantitative and qualitative job insecurity, as that relationship could partly explain the process of experiencing job insecurity over time. Particularly, this study attempted to bridge a gap in our knowledge by understanding the role that the experience of one dimension of job insecurity (i.e., quantitative job insecurity) might have in the process of experiencing the other aspect of job insecurity (i.e., qualitative job insecurity), and vice versa.

Job insecurity, either quantitative or qualitative, can be understood from a perspective of a perceived threat to work-related job resources, i.e., employment in general or a set of resources that being employed grants, respectively. Under the conservation of resources (COR) theory, these resources do not develop independently but rather form a collective [14]. That said, job resources expand or downgrade in the aggregate, which may suggest that two types of threats to job resources (i.e., quantitative and qualitative job insecurity) do not develop in isolation but rather relate to one another. In line with COR, we postulate that quantitative and qualitative job insecurity might be reciprocally related. Specifically, we consider resource caravan passageways as a set of conditions within which the rise of insecurities towards a particular job resource changes the environmental conditions for that employee and, ultimately, leads him/her to the perception of a threat to other job resources [15].

Let us illustrate how the relationship between quantitative and qualitative job insecurity might affect the continuous process of job insecurity with the following example. Let us picture two employees; employee “A” for some reason (the causes of the initial feeling of job insecurity are beyond the scope of this article), anticipates a higher-than-usual threat to job loss (i.e., quantitative job insecurity), whereas employee “B” anticipates a higher-than-usual threat to valued job features (i.e., qualitative job insecurity). If we follow these people over a specified period, we might observe that although objectively nothing has changed, employee “A” now also worries about the job characteristics while employee “B” feels more worried about job loss too. How is that possible? Well, employee “A”’s first thought was: “I might lose my job”. However, after some time of reflection (maybe a dinner conversation with a partner), they realize: “If I lose my job, I lose my salary, colleagues, social status, company car, etc., which are the goods that my job provides”. Thus, the initial threat of job loss leads to worries about the job features—qualitative job insecurity. In contrast, Employee “B” worries at first about keeping the job features. Maybe their company was going through changes (i.e., budget cuts), which led them to believe: “I will not get a bonus this year”. After a conversation with colleagues, they started generalizing their worries: “They might take my company car”, “Cut the salaries” or “Downgrade my position”. The budget cuts can even be perceived as signs of the upcoming dismissal of part of the workforce. Over time, employee “B” becomes exhausted with these worries—the negative thoughts and worries intensify, spread across, and finally, a worry about job loss takes over. Thus, in the instance of employee “B”, the initial threat to job characteristics leads to a threat of job loss—quantitative job insecurity.

The example above is undoubtedly a simplified version of the complex process of experiencing job insecurity. For one, employees usually feel, to some extent, both quantitative and qualitative job insecurity [16,17]. Thus, the two processes are most likely concurrent and interwoven. However, the example might help to envision the process that might occur within individual employees. Specifically, we argue that regardless of the initial cause, the sole appearance of either of these threats might begin a process of circular influence between one dimension of job insecurity (i.e., quantitative job insecurity) and the other (i.e., qualitative job insecurity), thus commencing a continuing process of job insecurity.

Although scholars agree that quantitative and qualitative job insecurity are related, the salience and direction of their relationship have not been extensively studied yet. Theretofore, research has cautiously suggested an underlying causal process, however, without sufficient evidence. For instance, a cluster of cross-sectional studies consistently suggests that employment, which is a key job resource, provides access to all other job resources (such as salary, career opportunities, healthcare, social status, etc.). Thus, quantitative job insecurities create conditions in which employees experience qualitative job insecurity (i.e., a threat of job loss poses a direct risk to all job resources) [18,19,20]. In contrast to these findings, a recent longitudinal study has found a reverse relationship, which suggested that over time a threat to important job features can be generalised towards a threat to the job as a whole [21]. Thus, so far, the conclusions regarding that relationship strongly diverge, and more research is utterly needed.

The contribution of this study is three-fold. First, we propose and empirically test a set of theory-driven hypotheses regarding the relationship between quantitative and qualitative job insecurity. We implement the conservation of resources (COR) theory to substantiate a research model that proposes a reciprocal relationship. In doing so, we perform a stepwise procedure to separately examine three viable ways in which quantitative and qualitative job insecurity might be related to one another. We begin with the most prevailing premise that experiencing a threat to job loss (i.e., quantitative job insecurity) leads to experiencing threats to job characteristics (i.e., qualitative job insecurity). Next, we test the reverse relationship and explore whether qualitative job insecurity leads to quantitative job insecurity. Finally, we test the joint process in which quantitative and qualitative job insecurity are reciprocally related. Second, we use a three-wave longitudinal design, which allows us to test the hypothesised effects using a repeated assessment of each participant (i.e., track individual changes). The results using longitudinal data tell us more about the time ordering of variables and consequently suggest the directionality of the associations [22]. Furthermore, in the current study, we chose the time interval between the observations to last 6 months. Although the optimal time lag for these associations is still unknown, past research has shown that the effects of both aspects on job insecurity are observable within 6 months [21,23,24,25,26]. Thus, a six-month time lag might be adequate to observe a rather more instant effect that a threat to job loss has on a threat to the loss of job characteristics and long enough for the reverse effects to develop [27].

Finally, in response to criticism that within-person processes are being wrongly estimated at the between-person level [28], we employ a person-centred approach. To date, the entire literature on the relationship between quantitative and qualitative job insecurity has been based on research that uses a variable approach, which assesses how a sample mean of quantitative job insecurity is associated with a sample mean of qualitative job insecurity. Accordingly, pasted research has failed to control for individual differences [29]. In other words, in the variable-centre approach, the results from employees who might be highly insecure and those who feel little to no insecurity are averaged into one value, which has been proven to give biased estimates and distort the judgement of the within-person processes [30]. Indeed, recent studies using a person-centred approach have indicated that the lion’s share of job insecurity variance (both quantitative and qualitative) is at the between-people level, i.e., employees significantly vary in their average experience of job insecurity [16,31,32]. Thereupon, to correctly estimate how the change in the experience of quantitative job insecurity relates to the change in qualitative job insecurity, we need to control individual differences. In the current study, we apply a statistical technique that acknowledges and properly distinguishes the between-person differences from the within-person variation. Specifically, we use a multi-indicator random intercept cross-lagged panel model (RI-CLPM), which is an extension of the traditional cross-lagged panel model (CLPM). In contrast to the traditional CLPM, RI-CLPM controls the stability of between-person differences through the inclusion of a random intercept. The random intercepts account for the between-person variance in quantitative and qualitative job insecurity so that the lagged relationships pertain to the within-person temporal changes in these two dimensions.

### 1.1. Job Insecurity

Job insecurity defines an individually perceived threat to the continuity of a current job situation [7]. As these perceptions can pertain to any change in employment, job insecurity is typically considered a two-dimensional construct. The first dimension, defined as quantitative job insecurity, refers to “the perceived threat of job loss and the worries related to that threat” [33]. In that respect, quantitative job insecurity encompasses the likelihood and worry of job loss in the nearest future [33]. The second dimension, defined as qualitative job insecurity, relates to a perceived threat to the continuity of important job features. Thus, qualitative job insecurity encompasses the likelihood and worry of losing valued job characteristics such as career prospects, wage stimulation, or the types of tasks embedded in the job description [8].

Undeniably, these two types of job insecurity share core characteristics. Firstly, both aspects are characterised by a perceived uncertainty about future employment; individuals do not know whether, and if so, how their job will continue or change. This experience comprises the perceived likelihood of a change to the current job situation and worries related to that threat [34]. Second, quantitative and qualitative job insecurity is related to anticipated involuntary changes to the job. Thus, the level of perceived job insecurity (either quantitative or qualitative) being reported is considered a discrepancy between the preferred level of job security [5,35], i.e., some employees might choose volatile work conditions and experience little to no insecurities. Consequently, the third characteristic is the personalised experience, which means that the employee’s subjective experience of threat is a key defining feature. In other words, the rise and consequences of job insecurity depend on how job security, via the anticipated workplace changes, is being perceived and appraised by individuals [5]. The working population is heterogenous, i.e., individuals are characterised by diverse demographics (i.e., age, gender, education level) and personality traits (i.e., self-esteem, negative affectivity, neuroticism, introversion, locus of control), which are common sources of job insecurity [10,36]. That said, the saliency of the experience of perceived job insecurity has been shown to differ even among employees whose workplace context is objectively comparable [37].

Bearing that in mind, it is to be expected that both dimensions are related. Indeed, research, which includes both aspects invariably, has found a significant positive correlation between quantitative and qualitative job insecurity [13,17,18,19,20,38,39,40]. Furthermore, comprehensive, person-centred analysis finds that both dimensions occur conjointly, even when controlling for sample heterogeneity. So far, three distinct profiles have been identified: (1) high job insecurity, for which both quantitative and qualitative job insecurities indicate elevated feelings of worry about the job situation; (2) low job insecurity, for which both quantitative and qualitative job insecurity indicates little to no worries; (3) moderate job insecurity, for which the employees experience some level of quantitative and qualitative job insecurity, but with distinguishably more prominent feelings of a threat to job characteristics [16,17].

Despite this clear alikeness, quantitative and qualitative job insecurity are conceptually different. Greenhalgh and Rosenblatt (1984), in their seminal article, defined job insecurity in broad terms as “a potential loss of continuity in a job situation” [7]. That definition inspired us to conceptualize job insecurity as a two-dimensional construct with two distinct aspects that separately measure a perceived threat to job loss as a whole and a perceived threat to valued job characteristics [35]. Two separate scales were constructed and validated to measure quantitative and qualitative job insecurity [13,41,42]. To date, confirmatory factor analysis (CFA) from all the past studies corroborates this conceptualisation, indicating that the quantitative and qualitative dimensions of job insecurity are separate variables [9,41].

Given that they are independent work stressors, the consequences of each dimension vary as well. Although both dimensions of job insecurity are linked with worsening health [28,29], work attitudes [30,31], performance [32], or behaviour [33], comparative research shows that the magnitude of these effects varies between the outcomes. For example, some studies have found that quantitative job insecurity was linked with mental and physical health, whilst qualitative job insecurity correlated stronger with work attitudes [8,21]. Furthermore, quantitative job insecurity was found to be associated with avoidance, whereas qualitative job insecurity affected learning goal orientations [34]. Therefore, employees’ well-being might be different when experiencing a threat of job loss compared to when they experience a high threat to job features. In addition, predictors of job insecurity might be different depending on the type of job insecurity. For instance, a permanent contract might protect you from experiencing a threat of job loss but not from a threat to job characteristics. In contrast, organisational practices and job conditions are more strongly associated with qualitative job insecurity than quantitative job insecurity. Consequently, employees with greater access to organisational resources (i.e., participation in the decision-making process, colleague/leader support, fairness, job autonomy) and lesser organisational demands (i.e., workload, role conflict, role ambiguity) are better protected from qualitative job insecurity but not necessarily from quantitative job insecurity [10,36].

### 1.2. Understanding the Relationship between Quantitative and Qualitative Job Insecurity

In line with the above, we consider quantitative and qualitative job insecurity as separate yet related to psychosocial work stressors. This raises the question of whether and how quantitative and qualitative job insecurity react to one another. In the following, we further elaborate on the plausible relationship between quantitative and qualitative job insecurity. Drawing on the conservation of resources (COR) theory, we explain the interrelationship between the two and conclusively substantiate a research model that proposes a reciprocal relationship.

On the one hand, the experience of high quantitative job insecurity could lead to an increase in the experience of qualitative job insecurity. Specifically, employees who worry about job loss are expected to experience a rise in threats to valued characteristics of that job [43,44]. Under the conservation of resources theory (COR), resources develop and exist in aggregates called resource caravans. In other words, resources (for example, in the work context) tend to emerge from common environmental and developmental conditions, which is why they all appear to relate to one another rather than being isolated entities [14]. In line with this, employment and specific work-related resources can be viewed as co-travelling resources aggregated in one job-specific resource caravan [45]. Employment, which is a key resource, grants access to a broad set of work-related resources. In that context, stress that originates from a threat to job loss (i.e., quantitative job insecurity) changes the environmental conditions surrounding job resources and downgrades the individuals’ perception regarding the sustainability of these resources. In other words, employees who perceive that they might lose their job might feel that the job features, such as salary, job position, social status, career development, etc., are exposed to the same threat. Consequently, quantitative job insecurity, which is a threat to employment, can be perceived as a direct threat to job characteristics while augmenting qualitative job insecurity.

Prior studies corroborate this view. First, in a longitudinal study, Selenko and Batinic [44] found that a threat of job loss was related to reduced financial benefits six months later. Expanding on these results, Vander Elst and colleagues [46] found that quantitative job insecurity was linked with an increase in threats to both manifest and latent work benefits six months later. These studies highlight the possible importance of quantitative job insecurity in shaping the experience of a threat to the conditions of the job. A year later, Chirumbolo and colleagues [18] integrated these findings and proposed the ‘job insecurity integrated model’, in which qualitative job insecurity mediated the effects of quantitative job insecurity on the outcomes. Using Jahoda’s deprivation theory, the authors cautiously suggested that since a loss of employment is synonymous with losing all job features, a threat of job loss might lead to a fear of losing job features. Although subsequent studies also found this association, they were all based on cross-sectional data, which is unfit to make statements regarding the ordering of the effects [19,20].

Following the theory and previous research, we expect that:

 **Hypothesis 1:**
*Quantitative job insecurity at t_y_ is positively associated with qualitative job insecurity at t_y+1_.*


On the other hand, it is plausible that this relationship is reversed. When individuals perceive their highly valued job characteristics to be threatened, over time, they might become more worried about the security of their whole employment. Continuing with the conservation of resources (COR) theory, employees who experience resource loss (or a threat) are more vulnerable to further loss (or a threat) and are less capable of resource gain [14]. In this respect, employees who perceive a threat to the conditions of their job might be more vulnerable to perceiving their employment as less secure. Prior research has shown that unproductive formal channels of communication lead employees to seek information through informal channels (i.e., gossip, rumours, urban legends, and casual conversations), which only intensifies feelings of insecurity [32]. Thereafter, when management fails to address rising threats toward the conditions in which the job is performed, these insecurities could spill over onto other job features and, overall, the continuity of employment itself. In times of ongoing organisational changes, in which job characteristics are expected to follow some reforms, employees might wonder whether these changes will affect only the officially communicated conditions of the job or whether they should be generalised—threatening other job resources—and, ultimately, the job itself. Consequently, we expect that over time a threat to job characteristics (i.e., qualitative job insecurity) will be generalised toward possible job loss (i.e., quantitative job insecurity).

Although theoretically plausible, empirical evidence for that relationship is limited, to date, only one study has examined this relationship. The findings suggest that qualitative job insecurity is associated with an increase in quantitative job insecurity six months later [21]. We expect that this association prevails at the within-person level. Thus, we propose the following:

 **Hypothesis 2:**
*Qualitative job insecurity t_y_ is positively associated with quantitative job insecurity, at t_y+1_.*


From this discussion, one can conclude that COR theory supports the relationship between quantitative and qualitative job insecurity in either direction. As mentioned above, job resources aggregate under one job-specific resource caravan, and they all might be related due to common environmental conditions. These common environmental conditions, also called resource caravan passageways, either foster and nurture or block and drain jointed resources. When an employee experiences an initial threat (either to job loss or job features), it deteriorates the conditions within the caravan passageway, which affects all job resources. Thus, via resource caravan passageways, initial loss (or threat) begets future loss. Accordingly, following Hobfoll’s COR theory, we suggest a third possibility that the two aspects of job insecurity affect each other reciprocally [14,47]. To evaluate the joint mechanism, we propose:

 **Hypothesis 3:**
*Quantitative and qualitative job insecurity are reciprocally related over time.*


### 1.3. Intrapersonal Approach to Job Insecurity

In the current study, we applied a person-centred approach to investigate the longitudinal relationship between quantitative and qualitative job insecurity over time. The rationale for choosing this method was twofold.

First, the person-centred approach controls for between-person differences, which is important since the experience of job insecurity differs across individuals rather than being homogeneous [48]. Past research that used a person-centred approach repeatedly showed that employees did not only differ in the intensity of the job insecurity experience (i.e., level of job insecurity) but also in the combination of two types of job insecurity (i.e., the shape of job insecurity) [16]. For instance, considering quantitative and qualitative job insecurity, Van Hootegem and colleagues [40] found five developmental patterns of job insecurity, which diverged between stable (high, moderate, low), decreasing, and increasing trends. Thus, across one sample, we might observe employees with a relatively stable low or a continuously high fear of future job loss next to employees who, over time, become either more or less secure about their job. Furthermore, constellations of job insecurity dimensions are distinguished between secure employees (low quantitative and qualitative job insecurity), employees who predominantly experience a threat to job characteristics (qualitative job insecurity dominant), and employees who feel generally insecure (high quantitative and qualitative job insecurity) [16,17]. Thus, employees do not only differ in the strengths and temporal stability of these threats but also in the combinations of the aspects that these threats represent. These findings are not surprising since, as we mentioned earlier, job insecurity is a subjectively perceived psychological stressor. In fact, personal resources have been found to be one of the most important predictors of job insecurity [10]. Thus, regardless of the objective changes in the work context, employees differ in their experience of job insecurity due to personal characteristics, such as personality traits (i.e., negative affectivity, locus of control, core self-evaluations, etc.) and the self-assessment of their skills and capabilities in the work domain (i.e., organisation-based self-esteem, employability, adaptation, etc.), which need to be controlled when exploring the processes that occur at the person-level [10,49,50].

To the best of our knowledge, past research has solely examined the relationship between quantitative and qualitative job insecurity at the between-person level, which captures the rank-order position of individuals. In that respect, cross-sectional studies found that employees who usually experienced a higher threat of job loss also reported a higher threat of job characteristics [18,19,20], whereas the longitudinal examination observed that employees who experienced higher qualitative job insecurity at one point in time, experienced higher threats to job loss six months later [21]. These are important findings that demonstrate how these two variables are related at the population level. Yet, they do not account for individual differences, which might lead to biased conclusions regarding associations between the two aspects of job insecurity [28,51].

Second, a person-centred approach explores the relationship between constructs at the person level [30]. The basic idea is that the observed variance of the construct— here—job insecurity—can be decomposed into stable differences between the individuals (How does this person feel when compared to how others feel?) and within-person fluctuations (How does this person feel when compared to how she/he usually feels?) [52]. In the current study, a core research question concerns the associations between two psychological constructs, which are processes that occur within a person [30,53]. Specifically, we want to know if a change (i.e., with-in-person fluctuations) in the experience of one aspect of job insecurity (i.e., quantitative job insecurity) at one point in time is related to a change (i.e., with-in-person fluctuations) in the experience of the other aspect of job insecurity (i.e., qualitative job insecurity) later on and vice versa. To answer this research question, we need to deconstruct the variance in job insecurity and, controlling for the between-person variance, analyse these associations at the within-person level; hence, we need to employ a person-centred approach.

### 1.4. Present Study

In congruence with the above, we proposed an empirical test of a theory-driven question regarding the relationship between quantitative and qualitative job insecurity. We implemented the conservation of resources (COR) theory to substantiate a research model that could propose a reciprocal relationship. Furthermore, we conducted a stepwise procedure to separately examine three viable ways in which quantitative and qualitative job insecurity might be related to one another. We begin with the most prevailing premise that experiencing a threat to job loss (quantitative job insecurity) could lead to experiencing a threat to job characteristics (qualitative job insecurity). Next, we tested the reverse relationship and explored whether qualitative job insecurity leads to quantitative job insecurity. Finally, we tested the joint process in which quantitative and qualitative job insecurity were reciprocally related.

To address our research question, we employed a person-centred approach using a three-wave longitudinal dataset. Given the core objective of this study, which was to examine the associations between the change in quantitative and qualitative job insecurity, we followed our participants over one year. An important issue that warrants attention in longitudinal research methodology is the appropriate time interval between the measurement observations. Specifically, a time length should correspond with the “real” time lag it takes for the effect to occur. If the chosen time lag is too short, then the predictor has insufficient time to affect the outcome. In contrast, if the chosen time lag is too long, the effect of the predictor on the outcome might already be too weak to detect [54]. To date, the optimal time lag with which to observe the associations between quantitative and qualitative job insecurity has not been established. However, past research examined the outcomes of job insecurity and successfully estimated the significant effects of both quantitative and qualitative job insecurity using a six-month time lag [21,23,24,25,40]. These findings suggest that a time lag of six months was adequate to observe the effects of one aspect of job insecurity on the other one and reverse. Consequently, we implemented a 6-month time lag. The data were analysed using a multiple indicator random intercept cross-lagged model (CLPM-RI) [51,55]. This method is superior to a traditional cross-lagged panel model as it controls for stable, trait-like differences between individuals. This implies that the lagged associations are calculated exclusively based on within-person fluctuations [55]. Specifically, the model splits the variance of each variable into a time-invariant, trait-like part (the individual’s average level of experienced quantitative and qualitative job insecurity; a between-person variance for quantitative and qualitative job insecurity—BQN_i_ and BQL_i_, respectively) and a time-varying, state-like part that captures dynamic, overtime fluctuations around the individual’s expected score (a within-person variance for quantitative and qualitative job insecurity—WQN_it_ and WQL_it_). The latter was used to estimate the autoregressive and cross-lagged effects, which tested the hypotheses formulated above.

## 2. Materials and Methods

### 2.1. Sample and Procedure

The current study used data collected from Belgian employees (the questionnaire was available in Dutch and in French to collect data from both Flemish and Walloon employees). The longitudinal design included three waves collected between January 2013 and May 2014 (a 6-month time lag between each measurement wave). Formerly, the data were collected for the research project on employability [56] and were shared by the authors for this study (The authors would like to thank Dr Jill Nelissen from KU Leuven for sharing their dataset and providing us with all the information about the data collection process. At the time of data collection there was no legal obligation to have the study approved by an independent ethical review board). Fifteen organisations were contacted, out of which thirteen agreed to participate in the survey. For all organisations, access to employees was facilitated through the Human Resource Department. For each wave, participants received two reminders to complete the questionnaire (online or on paper) at work or home. In the introduction to the survey, the researchers stated the purpose of the study and guaranteed voluntary participation and anonymous processing of the data. We sampled 4981 employees, of which 3694 participated in the first wave (response rate 74%). The first data collection (Time 1) took place between January and March 2013. The subsequent two waves occurred between October–November 2013 (Time 2) and April–May 2014 (Time 3). From the employees who participated in the first wave, we gathered 2045 employees who returned a questionnaire in the second wave (41% response rate) and 1698 employees who completed the survey in the third wave (34% response rate).

We excluded employees who did not fill in the questionnaire in the first wave (n = 1287), leaving us with a final sample of 3694 participants; 42% were women (n = 1539), 52% were men (n = 1901) and 6% (n = 254) left the question without an answer. The average age was 41.86 years (SD = 10.48), and almost half of the participants had a degree in higher education (49% in comparison to 21% of participants with a middle level of education and 20% with a low level of education). Employees with a permanent job contract dominated the sample (n = 3322, 90%), and 80% (n = 2957) worked full-time. On average, employees had worked 11.9 years (SD = 10.6) in the same organisation and 7.41 years (SD = 8.25) in the same position. Respectively, 26% (n = 951) were blue-collar workers (8% unskilled labourers and 18% skilled workers), 36% (n = 1324) were white-collar employees (17% lower-level and administrative clerk; 19% middle-level employee), and 31% (n = 1152) had a managerial position (23% low- and middle-level management; 8% senior management). Respondents worked across 13 different organisations; 17% of the sample (n = 635) worked in two companies from an industrial sector (secondary sector), 41% (n = 1523) worked in six organisations from a service sector (tertiary sector), and 41% (n = 1520) worked in five organisations from a public sector (quaternary sector). Finally, the respondents were asked whether they experienced a change in their work context in the six months before the study. Almost three-quarters of respondents (67.1%; n = 2478) did not experience any change to their job; 10% changed position (n = 369); 7.6% (n = 280) changed job level; and less than 2.5% experienced a change in the workplace (21 respondents changed their team, 48 respondents changed from employer, and 20 respondents changed a sector). Thus, the majority of respondents stayed at the same job throughout the observation period. We can conclude that the sample was heterogeneous regarding its composition.

### 2.2. Drop-Out Analysis

We analysed possible attrition bias using multinomial logistic regression. We included study variables at Time 1 (quantitative and qualitative job insecurity) and background variables: work time frame (0—full time, 1—part-time), type of contract (0—permanent contract, 1—temporary contract), work experience (years), tenure (organisational and positional; years), gender (0—women, 1—men), age (years), education (0—low education, 1—middle education, 2—high education), and position (0—blue collar workers, 1—white collar workers, 2—management). The results indicate that the odds of dropping out after the first wave (T1) vs. participating in all three waves (T1T2T3) increased by 1.662 when moving from low education to middle education and 1.477 when moving from low education to high. In contrast, the odds of dropping out after the first wave (T1) vs. participating in all three waves (T1T2T3) decreased by 0.301 when moving from blue-collar to white-collar and 0.384 when moving from blue-collar to management. In other words, people who dropped out after the first wave were more likely to present middle and high education and work in blue-collar positions. They might represent a group of employees who, despite high education, landed low-skilled jobs. Furthermore, the odds of not participating in the second wave (T1T3) decreased by 0.7 when moving from women to men, which means that women were more likely to drop out during the second wave. To limit the bias associated with a systematic drop-out, we used a full information likelihood estimation (FIML), which uses partially incomplete data by estimating only the parameters of those variables that were observed for that individual [57]. FIML estimates were found to be unbiased and more efficient than any other method when handling missing data (listwise deletion, pairwise deletion, imputations) under MAR and MCAR mechanisms [58,59].

### 2.3. Measurements

**Quantitative job insecurity.** Quantitative job insecurity was measured with the four-item scale developed by De Witte [60] and validated by Vander Elst and colleagues [42]. It measures the perceived likelihood (e.g., “There is a chance that I will soon lose my job”) and worries about job loss (e.g., “I feel insecure about the future of my job”). The items were rated on a five-point Likert scale from 1 (totally disagree) to 5 (totally agree). The internal consistency for the current sample was α = 0.85 at T1, α = 0.86 at T2, and α = 0.87 at T3.

**Qualitative job insecurity**. Qualitative job insecurity was measured with a four-item scale developed by De Witte and De Cuyper and recently validated by Fischmann and colleagues [41]. It measures the perceived likelihood (e.g., “There is a chance that my job will change in a negative way”) and worry of loss or negative changes in the overall job content and working conditions (e.g., “I worry about what my job will look like in the future”). The items were rated on a five-point Likert scale from 1 (totally disagree) to 5 (totally agree). The internal consistency for the current sample was α = 0.91 at T1, α = 0.92 at T2, and α = 0.92 at T3.

**Descriptive variables.** Background data were included just to visualise how the development of quantitative and qualitative job insecurity varied across the groups, as the RI-CLPM controlled the stable between-person differences. We included information about the participants that are commonly used as control variables: gender (0—female; 1—male), education (0—low education; 1—middle education; 2—high education), contract (0—permanent; 1—temporary), work time frame (0—full-time; 1—part-time), organisational tenure (0—up to 5 years; 1—up to 15 years; 2—above 16 years), and positional tenure (0—below 1 year; 1—up to 5 years; 2—more than 5 years).

### 2.4. Analysis

Table 1 shows the means, standard deviations, and correlations for quantitative and qualitative job insecurity and background variables. Across the sample, the level of qualitative job insecurity was higher than that of quantitative job insecurity, which meant that across all three waves, on average, employees experienced a higher threat to job characteristics than a threat of job loss. Furthermore, the means of quantitative and qualitative job insecurity were invariant across the observation period. In other words, on average, the sample experienced a continuous similar level of job insecurity. Although bivariate correlations found no significant differences in quantitative and qualitative job insecurity for various positions (blue-collar, white-collar, management) and time frames (full-time, part-time), there were significant differences between groups with different types of contracts, work experience, tenure, education, age, and gender. Indeed, the graphical representation of the development of quantitative and qualitative job insecurity across time for each group represents just how much variance there is between the respondents (see Appendix A). For example, employees with a permanent contract experienced, on average, a low and stable threat to job loss. In comparison, employees with a temporary contract experienced a high threat of job loss with a decreasing tendency over time. Similarly, employees with short organisational (up to 5 years) and positional (up to 1 year) tenure experienced a sharp increase in threats to job characteristics, which stabilised for employees with a higher tenure.

These results suggest significant variability at the between-person level for both quantitative and qualitative job insecurity. To test this assumption, we explored the amount of variance that could be explained by stable trait-like differences between people (interindividual differences) vs. within-person fluctuations (intraindividual change). We used the reliability-adjusted intraclass correlation coefficient ICC(1) to account for the measurement error of quantitative and qualitative job insecurity. The measurement error has been shown to induce bias in the estimation of the ICC by increasing the within-person variance [61]. The reliability-adjusted ICC(1) weights the within-person variance with the construct’s reliability, which has been proven to result in robust estimates of ICC(1) [61,62]. The adjusted ICC(1) for quantitative job insecurity was 0.67, indicating that 67% of the variance could be explained at the between-person level (stable-trait, interindividual differences), while the remaining 33% was a within-person fluctuation (over time, intraindividual change). Similarly, the ICC(1) for qualitative job insecurity indicated that 63% of the variance could be explained by the between-person differences (37% by a within-person fluctuation).

For data analysis, we conducted structural equation modelling using the Lavaan package in R software [63]. We followed the instructions specified by Mulder and Hamaker [55] (for a similar methodology, see [31,64,65]). We used full-information maximum likelihood (FIML) to handle the missing data. The model fit was evaluated using several goodness-of-fit indices: (a) a comparative fit index (CFI), (b) the Tucker-Lewis index (TLI), (c) the root mean square error of approximation (RMSEA), and (d) the standardised root mean square residual (SRMR) [66,67]. A good model fit was indicated with CFI and TLI values equal to 0.95 or higher, RMSEA and SRMR with values of 0.6 and 0.8 or lower, respectively [68]. Alternative models were compared based on ΔCFI and ΔRMSEA, where a change in ≤−0.01 and ≤0.015, respectively, indicated a better model fit [69,70].

The analysis consisted of three steps. In the first step, we performed confirmatory factor analysis (CFA) [66]. We compared the hypothesised two-factor measurement model (M1) to: (a) a one-factor model (M2), which measured job insecurity as one general latent variable, and (b) a four-factor model (M3), in which quantitative and qualitative job insecurity dimensions were further split into a cognitive and an affective subdimension. In each model, the measurement errors were set to covary across time.

In the second step, we assessed a longitudinal measurement invariance to test how well the measured items represented the underlying latent constructs across time [71]. We compared a set of nested models, where each model represented a more rigid invariance than the previous model. We started with the configural invariance model, i.e., an unconstrained model with equal factor structure across time, as our baseline model. Next, we estimated the metric invariance model (M4), which placed equality constraints on factor loadings of the corresponding items across time. The strong invariance model (M5) added equality constraints to the items’ intercepts. Finally, the strict invariance model (M6) constrained residual variances. Mulder and Hamaker [55] indicated that metric invariance was a minimum requirement to specify the RI-CLPM and evaluate the structural paths at the within-person level.

In the final step, we estimated the random intercepts cross-lagged panel model (RI-CLPM). We decomposed estimated latent variables to (a) random intercepts, which accounted for stable differences in the mean levels of quantitative and qualitative job insecurity between the employees, and (b) a within-person component, which is the intraindividual variation around the individual’s average level, across time. The structural model, which contains autoregressive and lagged paths, was added at the within-person level (M7). Specifically, we estimated the autoregressive paths (i.e., the extent to which a within-person deviation from the expected score at time t can be predicted by a within-person deviation from the expected score at time t − 1) and cross-lagged paths (i.e., the extent to which the within-person fluctuation in qualitative job insecurity at time t is predicted by the within-person fluctuation in quantitative job insecurity at time t − 1, and reverse). Finally, we examined whether the lagged effect remained stable over time. We compared three models, in which we added equality constraints on the autoregressive paths (M8), the cross-lagged paths from qualitative job insecurity to quantitative job insecurity (paths a; M9), and the cross-lagged paths from quantitative job insecurity to qualitative job insecurity (paths b; M10). The hypotheses were tested on the model, which ultimately had the best fit for the data.

## 3. Results

### 3.1. Measurement Model and Longitudinal Measurement Invariance

Table 2 presents the results of the CFA and subsequent evaluation of the longitudinal measurement invariance. The hypothesised two-factor model (M1), which separately measures quantitative and qualitative job insecurity at each time point, showed a good model fit (χ^2^ (213) = 1994.632, CFI = 0.961, TLI = 0.949, RMSEA = 0.048, SRMR = 0.051). We compared that model with two alternative models. The first, one-factor model showed a poor fit to the data (χ^2^ (225) = 9382.963, CFI = 0.799, TLI = 0.753, RMSEA = 0.105, SRMR = 0.108), and the hypothesised 2-factor model showed significantly better fit (Δ χ2(12) = 7388.331, *p* < 0.001). The second, four-factor model (M3), did not converge, which meant that the quantitative and qualitative job insecurity could not be separated into cognitive and affective subdimensions. Therefore, the hypothesised 2-factor model was chosen as the baseline model. Next, we examined the measurement invariance. Gradually, the added equality constraints did not decrease the model fit. The strict measurement invariant model (M7) showed a satisfactory model fit (χ^2^ (257) = 2259.832, CFI = 0.956, TLI = 0.953, RMSEA = 0.046, SRMR = 0.053) and met the measurement invariance criteria (ΔCFI = 0.002, ΔRMSEA = 0). Hence, the measurement model with a strict measurement invariance was used to estimate the RI-CLPM.

### 3.2. Test of the Hypotheses: RI-CLPM and Stability of the Model

The RI-CLPM showed good fit to the data (χ^2^ (254) = 2218.475, CFI = 0.957, TLI = 0.953, RMSEA = 0.046, SRMR = 0.052). Accordingly, we examined the stability of the structural model. The model with equality constraints on the autoregressive paths (M8) did not worsen the model fit (ΔCFI = 0, ΔRMSEA = 0). Similarly, the additional constraints on the lagged paths from qualitative job insecurity to quantitative job insecurity (M9) did not compromise the model fit (ΔCFI = 0, ΔRMSEA = 0.001). The final model (M10) with constraints on all structural paths showed a good model fit (χ^2^ (258) = 2221.255, CFI = 0.957, TLI = 0.954, RMSEA = 0.045, SRMR = 0.052) and was not significantly worse than the partially constrained model (ΔCFI = 0, ΔRMSEA = 0). Table 3 provides an overview of the results. We concluded that the relationship between the constructs was invariant across time, and we proceeded to examine the hypotheses based on the final model.

We interpreted the results from the RI-CLPM as follows: (1) the results at the between-person level, (2) the cross-sectional covariation at T1 and the residual covariation at T2 and T3, (3) the autoregressive paths, and (4) the cross-lagged paths. The standardised coefficients of the final model are graphically depicted in Figure 1.

First, at the between-person level, the random intercept for quantitative job insecurity correlated positively with the random intercept for qualitative job insecurity (β = 0.261, *p* < 0.001), which meant that employees with a threat of job loss which was higher than the sample average also experienced a higher threat to job characteristics.

Next, we moved to the estimates at the with-person level. Cross-sectional covariance analysis showed significant T1 covariation (β = 0.160, *p* < 0.001) and T2, and T3 residual covariation (β = 0.198, *p* < 0.001; β = 0.161, *p* < 0.001, respectively). These results indicate that a within-person change (deviation from the individual’s expected score) in quantitative job insecurity is positively associated with a within-person change in qualitative job insecurity. In other words, employees who experienced a higher-than-expected level of quantitative job insecurity at one point in time, also reported a spike in the experience of qualitative job insecurity.

Third, the autoregressive paths were significant for quantitative job insecurity (β = 0.359 *p* < 0.001) but not for qualitative job insecurity (β = −0.111, *p* = 0.074). These results suggest that employees who at one point in time experienced an increase in the threat of job loss (t − 1) were more likely to experience an increase in the threat of job loss later in time (t).

Finally, we analysed the intraindividual cross-lagged paths between quantitative and qualitative job insecurity. In line with hypothesis 1, quantitative job insecurity, was positively associated with qualitative job insecurity, six months later (β = 0.238, *p* < 0.001). Employees who experienced a higher-than-usual threat of job loss at one point in time experienced a higher-than-usual threat to job characteristics six months later. In contrast with hypothesis 2, the results did not support a positive lagged association between qualitative job insecurity and quantitative job insecurity. These findings led to the rejection of hypothesis 3, which suggested a reciprocal relationship between quantitative and qualitative job insecurity.

## 4. Discussion

The objective of this project was to identify the relationship between quantitative and qualitative job insecurity. We implemented the conservation of resources (COR) theory to propose a reciprocal relationship. Furthermore, we conducted a stepwise procedure to separately examine each direction in which quantitative and qualitative job insecurity might be related to one another. To answer our research question, we performed a multiple indicator random-intercept cross-lagged panel model (RI-CLPM) [55]. We used three-wave longitudinal data collected every 6 months from the Belgian-employed population. The results were consistent with prior research, as we found a positive and significant correlation between quantitative and qualitative job insecurity at the between-person level. That said, employees who, on average, reported a higher level of quantitative job insecurity were more likely, on average, to report a higher level of qualitative job insecurity. In addition, quantitative and qualitative job insecurity were found to be significantly related at the within-person level, which indicated a positive interaction between an experienced change in quantitative and qualitative job insecurity. In other words, an employee who at t_y_ experienced a higher-than-usual (or a lower-than-usual) threat to job loss was more likely to simultaneously experience a higher-than-usual (or a lower-than-usual) threat to job features. Lastly, the cross-lagged analysis of the relationship between quantitative and qualitative job insecurity failed to confirm a reciprocal relationship. The results indicate a unidirectional relationship. Specifically, we found that employees who experienced a higher-than-usual threat of job loss were prone to experiencing a higher-than-usual threat to job characteristics six months later [52].

### 4.1. Theoretical Implications

First, we looked at the concurrent interdependence between quantitative and qualitative job insecurity. At the between-person level, the results showed a significant positive association between quantitative and qualitative job insecurity, which meant that employees who, on average, experienced a higher threat of job loss compared to other employees also tended to experience a higher threat to job characteristics. An analogous association was found at each measurement point at the within-person level. Specifically, employees who experienced an increase (or decrease) in quantitative job insecurity (from their average score) simultaneously experienced an increase (or decrease) in qualitative job insecurity. These results indicate that, although both dimensions of job insecurity are independent stressors, they are strongly related to one another at the baseline (average experience of quantitative and qualitative job insecurity) and in the trajectory of their changes over time.

These observed interdependencies between quantitative and qualitative job insecurity could be linked with the sources of job insecurity. An abundance of personal and environmental factors has been linked with job insecurity [5]. These variables could be further classified as stable or time-invariant such as demographics or personality traits, and more dynamic (time-varying) organisational changes. Although it has never been explicitly tested, we used our results to cautiously propose that the link between the quantitative and qualitative job insecurity at the between-person level (the baseline) is due to the stable sources of job insecurity (personal characteristics and demographics), whereas the link between the time-dependent discrepancies (over time fluctuations from a mean) is due to more volatile organisational changes.

In a recent meta-analytical study, Jiang and colleagues [10] synthesised previous studies and concluded that personality characteristics (i.e., positive vs. negative affectivity, neuroticism, internal vs. external locus of control, extraversion vs. introversion, secure vs. insecure attachment) are important predictors of felt job insecurity and were found to have a similar effect on how employees experienced both, a threat of job loss and a threat to job characteristics. For instance, employees with higher (vs. lower) levels of an internal locus of control appeared to experience lower job insecurity [50], and this association seemed comparable for both dimensions [72]. Similarly, demographic variables such as tenure, gender, educational level, union membership, employee contract, and occupational position, which are often included as control variables, are also associated with experiencing quantitative and qualitative job insecurity [36]. Looking at the distribution of job insecurity across our sample (see Appendix A), we indeed observed a tendency for quantitative and qualitative dimensions to proceed hand in hand (higher levels of quantitative and qualitative job insecurity for women vs. men, part-time vs. full-time, white-collar vs. blue-collar, etc.). These variables are stable (time-invariant), which suggests that their effect on quantitative and qualitative job insecurity remains constant for each respondent over time. In that respect, we propose that these variables (time-invariant antecedents) directly affect quantitative and qualitative job insecurity at the between-person variance rather than at the within-person level.

At the same time, organisational factors are more volatile—hence, they might better explain within-person, overtime fluctuations of quantitative and qualitative job insecurity. Organisational factors could be defined as current work conditions that either enhance employees’ well-being or cause strains [73]. Jiang and colleagues [10] argued that employees who have greater access to structural resources (i.e., more job autonomy, participation in decision-making, or greater organisational communication) or social resources at their workplace (i.e., peer support, organisational trust, good relationship with the supervisor), are likely to report lower than usual levels of job insecurity. In contrast, employees who experience higher organisational demands (i.e., work pressure, workload, conflicts, organisational change, or abusive supervision) or a decreased availability of resources are expected to report a spike in the experience of threats to job loss and job characteristics. In that respect, as job resources and demands fluctuate, the effect they have on an individual’s perception of job insecurity varies too. Thus, we proposed that organisational changes (time-variant antecedents of job insecurity) are directly linked with quantitative and qualitative job insecurity at the within-person level rather than at the between-person level.

To our knowledge, this is the first study that has examined the associations between the two dimensions of job insecurity, separately at within and between-person levels. Our results are in line with the previous studies, which suggest that quantitative and qualitative job insecurity are similarly affected by antecedents [10,36]. We advance a step further and cautiously suggest that the stability of the work-related variables (time-invariant vs. time-variant) determines whether they affect the average perception of job insecurity (between-person variance) or its time-depending fluctuations (within-person). It is, however, only an interpretation of the results and an attempt to link our results with the current knowledge of antecedents. Future research should evaluate this explicitly.

Next, the intraindividual autoregressive paths were significant for quantitative job insecurity but not for qualitative job insecurity. The carry-over effect of quantitative job insecurity was positive, meaning that an individual who experienced a higher threat of job loss (concerning the individual’s average level) was more likely to continue experiencing their job as insecure six months later. On the other hand, the carry-over effect of qualitative job insecurity was nonsignificant, which meant that a higher perception of qualitative job insecurity (concerning individuals’ average) at one point did not predict an elevated perception of this threat in the future. This aligns with the previous research that found nonsignificant within-person stability effects on qualitative job insecurity [31].

One explanation for these results may be related to how we define and operationalise quantitative versus qualitative job insecurity. Specifically, quantitative job insecurity is a perceived threat to one resource—employment—and, therefore, it only measures the worry and likelihood of job loss [74]. On the other hand, qualitative job insecurity is defined and measured as a threat to many resources (unspecified job characteristics), which aims to cover an abundance of job features [41]. When studied longitudinally, it could be that for the same employee, the measurement of qualitative job insecurity means a perceived threat of decreased salary at one time (measurement at time 1) and decreased career opportunities next (for example, here six months or a year later measured at time 2 or 3). Consequently, previous experience measured with a qualitative job insecurity scale was less accurate in predicting the current threat to job characteristics compared with the predictive power of a quantitative job insecurity scale. It is important to bear in mind that this is the first study to conduct these analyses at the within-person level. Thus, we cautiously suggest that the operationalisation of the tool might play a role in explaining the results. We further encourage more person-centred studies to better understand the role and the consequences of the within-person carry-over effects.

The core interest of this study was on the within-person, cross-lagged effects to test how quantitative and qualitative job insecurity predict one another over time. We found that a temporal deviation in quantitative job insecurity was positively associated with a temporal deviation in qualitative job insecurity six months later. These results indicate that an employee who experiences a higher-than-usual threat of job loss is more likely to experience a higher-than-usual threat to job characteristics six months later. This supports our hypothesis that job-related resources travel in aggregates, called resource caravans, and that a threat to the main job resource—employment—poses a direct threat to all job-related resources, which ultimately makes employees worried about losing important job features. Furthermore, these effects can be further supported by Jahoda’s deprivation theory. According to Jahoda’s deprivation theory, employment grants access to unique work resources, such as financial stability, social status, time structure, life purpose, career goals, and daily activities [75]. As such, losing a job (or, in this case, the anticipation of job loss) triggers the loss (or threat to loss) of all benefits (resources) that the job provides. Our results are in line with the prediction derived from Chirumbolo’s ‘job insecurity integrated model’ [18]. In addition, the current study replicates and expands on the previous cross-sectional studies in this field, giving empirical evidence on the ordering of variables, thus providing evidence for plausible causality effects [18,19,20].

In contrast, the reverse relationship was not found. Individuals who experienced a higher-than-usual threat to job characteristics did not experience a higher-than-usual threat to job loss six months later. This outcome contradicts that of Nawrocka and colleagues [21], who found a positive longitudinal association between qualitative job insecurity and quantitative job insecurity. This inconsistency could be attributed to a different method used in this study. Specifically, a RI-CLPM differentiates between-person effects from within-person effects, while the traditional CLPM, used in the previous research, does not. By controlling the between-person differences and by analysing the relationship between quantitative and qualitative job insecurity at the individual level, RI-CLPM might provide better estimates of these within-person processes. To check whether the method of the analysis affects the actual results in this study, we additionally conducted a traditional CLPM and compared the results with those of the RI-CLPM. The model fit for RI-CLPM was significantly better than for CLPM (Δ χ2(8) = 905.217, *p* < 0.001). The standardised coefficients of the RI-CLPM and traditional CLPM were different in significance levels and effect size for almost all associations. For example, the autoregressive paths for quantitative job insecurity in the CLPM were almost double the size when compared with the results from RI-CLPM (β = 0.788 and β = 0.808 vs. β = 0.427 and β = 0.470). Additionally, when measured with CLPM, the autoregressive paths for qualitative job insecurity were positive, whereas the results of the RI-CLPM were nonsignificant. As for the cross-lagged effects, CLPM indicated a stronger effect size from quantitative job insecurity to qualitative job insecurity than RI-CLPM (β = 0.355 and β = 0.322 vs. β = 0.152 and β = 0.156). Despite these differences, CLPM, similarly to RI-CLPM, found no significant overtime associations between qualitative job insecurity and quantitative job insecurity (β = −0.017 vs. β = −0.094 and β = −0.096). Thus, for this sample, we found that qualitative job insecurity did not predict quantitative job insecurity when tested with RI-CLPM, as well as with a traditional CLPM. However, we still call for a cautious interpretation of these results, as this is the first study to examine these associations at the within-person level. To develop a full picture of whether qualitative job insecurity predicts quantitative job insecurity or not, further longitudinal research is needed.

Bringing these results together, we found no evidence for an intraindividual cycle between quantitative and qualitative job insecurity. However, as this is only the second study that examines the reciprocal relationship, the evidence is currently too scarce to make a firm conclusion. It is somewhat surprising that our results were in contrast with the previous study, which found that qualitative job insecurity predicted quantitative job insecurity six months later [21]. However, we could argue that these inconsistencies might support our idea that the relationship between quantitative and qualitative job insecurity is, in fact, bidirectional; a 6 months time lag could be a breaking point for that cycle.

It is expected that a threat of job loss has a rather instant effect on a threat to job characteristics (i.e., if I perceive my job as insecure, I will rather quickly perceive my work benefits to be less secure). In contrast, a threat to job characteristics might need to take time to directly affect a threat of job loss (i.e., if I perceive my career opportunities hampered, I will not immediately fear for my job. However, with time, this threat might eventually lead to an increase in worry for the employment). Considering the 6 months time lag used in both studies, the current study might have caught the last moments of the effects that a threat of job loss has on a threat to job characteristics, whereas in the previous study by Nawrocka and colleagues [21], this effect might have been already imperceptible, while the effect of the threat to job characteristics on a threat of job loss has gained power and emerged in the analysis.

To date, the optimal time interval to test this relationship remains unclear, as only a 6 months time lag between quantitative and qualitative job insecurity has been analysed. Studies on time lags suggest that effects are more prominent when shorter time lags are applied [76]. Thus, the reciprocal effect might be observed with a time lag, that is, shorter than 6 months. On the other hand, some scholars argue that weak effects need a longer time to unfold [27]. This might explain why the effects of qualitative job insecurity on quantitative job insecurity were not present. In other words, quantitative job insecurity, which is a threat to overall employment, might have a strong and rapid effect on qualitative job insecurity, whereas threats to job characteristics, as a milder type of threat, might take a longer time to unfold and affect employee’s perceived overall job security. One way to estimate an optimal time lag could be through a “shortitudinal” pilot study, which is conducted with time intervals that are most likely shorter than the optimal time lag and then estimates how various time lags affect the association between the two dimensions of job insecurity [27].

### 4.2. Limitations and Future Research

As with any study, a myriad of limitations should be considered. First, we used self-reported measures, which could raise concerns about common method bias and response bias (i.e., social desirability). We tried to decrease the risk of these biases by (1) highlighting voluntary and anonymous participation, (2) using internationally validated scales, and (3) separating in time the predictor and the outcomes variables [77]. Furthermore, the data were collected independently, without any involvement from the companies’ management, which decreased the risk of socially desirable answers.

Second, the data were collected via a non-probability sampling procedure, which might have resulted in a sampling bias. More specifically, the data were collected from 14 organisations; hence, certain working groups lacked proper representation. In addition, women were slightly underrepresented, while full-time workers were overrepresented compared to the Belgian population [78]. Additionally, from the dropout analysis, we found that dropout was more likely among higher-educated employees and blue-collar workers. Overall, we dealt with a sample that did not correctly represent the population of Belgian employees when considering demographic variables. However, in their meta-analytical study, Jiang and colleagues [10] argued that demographic variables are poor predictors of job insecurity. Furthermore, via RI-CLPM, we controlled these individual differences. Thus, higher dropout amongst employees with higher education and blue-collar workers and an inaccurate representation of the Belgian working population should not have significantly influenced our results.

Finally, although we controlled for between-person differences, there might be heterogeneity at the within-person level that we did not account for in the current study. Although RI-CLPM allowed us to control for between-person differences, it assumed that individual responses to the temporal deviations were identical. In other words, we expected all employees who experienced higher than their average threat of job loss to report higher than their average threat to job characteristics six months later and vice versa. However, the relationship between quantitative and qualitative job insecurity may be different between groups of individuals who differ according to certain characteristics. Accordingly, we used a set of demographic variables (type of contract, work time frame, work experience, organisational and positional tenure, position, education level, and gender) and conducted a multi-group RI-CLPM to test whether within-person processes differed between these groups of individuals (detailed results are available upon the request) [55]. Using chi-square difference testing, we compared a model where lagged regressions were freely estimated for each group (for example, employees with temporary vs. permanent contracts) with a model including equality constraints on these parameters across the groups. The results revealed that all models with equality constraints fit the data better, which indicated that the within-person processes were similar regardless of their background characteristics. Nevertheless, the relationship between quantitative and qualitative job insecurity might be different for employees who differed in the variables that were not measured in this study. Next to demographic variables, which were unmeasured in the current study, third variables, such as job resources and job demands, were found as strong predictors of job insecurity [10]. Thus, future studies could test if and how the relationship between quantitative and qualitative job insecurity differs for groups of employees, conditional on their access to work-related resources and/or the intensity of present job demands.

## 5. Conclusions

Overall, our results highlighted the interdependence between quantitative and qualitative job insecurity, separately at the between-person (employees who usually feel insecure about their job also feel insecure about their job characteristics) and the within-person level (an employee who experiences an increase in threat to job loss concurrently experience higher than their usual threat to job characteristics). Moreover, with the results that indicate over 60% of the variance in both quantitative and qualitative job insecurity present at the between-person level, we call for a determined shift towards person-oriented research in the field of job insecurity. Although the empirical evidence did not prove the expected reciprocal relationship, it gave longitudinal support for Chirumbolo’s ‘job insecurity integrated model’ [18]. More specifically, our findings suggest that an employee who experiences a higher-than-usual threat of job loss (quantitative job insecurity) is more likely to experience a higher-than-usual threat to job characteristics (qualitative job insecurity) six months later.

## Figures and Tables

**Figure 1 ijerph-20-05280-f001:**
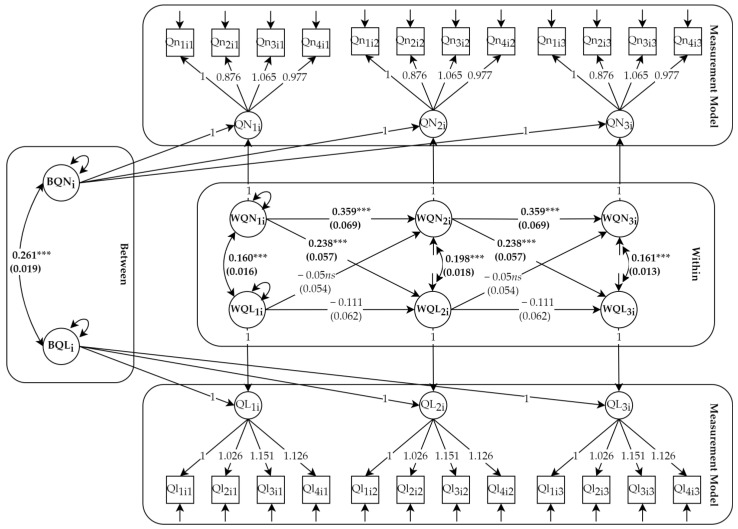
The results of the random-intercepts cross-lagged panel model with standardised path coefficients. Note: *** *p* < 0.001; QNti—latent quantitative job insecurity; QLti—latent qualitative job insecurity; BQNi—random intercept for quantitative job insecurity; BQLi—random intercept for qualitative job insecurity; WQNti—within-person fluctuations in quantitative job insecurity; WQLti—within-person fluctuations in qualitative job insecurity. The graphical representation of the research model is based on the article by Mulder and Hamaker [55].

**Table 1 ijerph-20-05280-t001:** Means, standard deviations, reliability (Cronbach’s alpha in parentheses), and correlations.

		M	SD	1	2	3	4	5	6	7	8	9	10	11	12	13	14	15
1	Quan T1	2.22	0.87	(0.85)														
2	Quan T2	2.29	0.88	0.65 ***	(0.86)													
3	Quan T3	2.21	0.86	0.58 ***	0.66 ***	(0.87)												
4	Qual T1	2.52	1.03	0.54 ***	0.40 ***	0.36 ***	(0.91)											
5	Qual T2	2.57	1.00	0.41 ***	0.61 ***	0.41 ***	0.61 ***	(0.92)										
6	Qual T3	2.58	1.03	0.37 ***	0.41 ***	0.57 ***	0.60 ***	0.63 ***	(0.92)									
7	Time Frame	1.20	0.40	0.05 **	0.03	0.03	0.06 ***	0.03	0.04	─								
8	Contract	1.10	0.29	0.27 ***	0.18 ***	0.20 ***	0.02	0.02	0.05	0.05 **	─							
9	Position	3.59	1.46	−0.01	−0.01	0.00	0.03	0.01	0.02	−0.01	0.07 ***	─						
10	Work Exp (Years)	18.76	11.15	−0.08 ***	−0.12 ***	−0.11 ***	−0.01	−0.05 *	−0.08 **	0.10 ***	−0.23 ***	−0.05 *	─					
11	Org. Tenure (Years)	11.89	10.60	−0.11 ***	−0.12 ***	−0.13 ***	0.06 ***	0.02	−0.03	0.09 ***	−0.25 ***	0.05 **	0.71 ***	─				
12	Pos. Tenure (Years)	7.42	8.25	−0.14 ***	−0.17 ***	−0.15 ***	0.05 **	0.02	0.03	0.10 ***	−0.18 ***	−0.14 ***	0.51 ***	0.63 ***	─			
13	Gender	1.55	0.50	−0.05 **	−0.06 **	−0.07 **	−0.06 ***	−0.06 **	−0.05	−0.33 ***	−0.10 ***	−0.20 ***	0.20 ***	0.11 ***	0.08 ***	─		
14	Age	41.87	10.47	−0.07 ***	−0.12 ***	−0.10 ***	0.00	−0.04	−0.07 **	0.14 ***	−0.20 ***	0.02	0.93 ***	0.68 ***	0.50 ***	0.14 ***	─	
15	Education	7.86	2.85	0.04 *	0.07 **	0.07 **	0.10 ***	0.10 ***	0.11 ***	0.03	0.13 ***	0.71 ***	−0.29 ***	−0.14 ***	−0.20 ***	−0.31 ***	−0.22 ***	─

Note: N = 3694. Quan T1–T3—score for quantitative job insecurity at times 1, 2 and 3; Qual T1–T3—score for qualitative job insecurity at times 1, 2 and 3; Full vs. Part-time: 1—full-time, 2—part-time; Contract: 1—permanent, 2—temporary; Position: 1—unskilled blue-collar, 2—skilled blue-collar, 3—administrative clerk, 4—middle-level employee, 5—lower and middle-level management, 6—senior management; Gender: 1—women, 2—men; Education: 1—no degree, 2—primary education, 3—lower secondary vocational education, 4—lower secondary technical education, 5—lower secondary general education, 6—higher secondary vocational education, 7—higher secondary technical education, 8—higher secondary general education, 9—higher education (professional bachelor), 10—higher education (licentiate/master), 11—university education (master), 12—doctorate; * *p* < 0.05; ** *p* < 0.01; *** *p* < 0.001.

**Table 2 ijerph-20-05280-t002:** Fit indices of competing nested factor models and standardised maximum likelihood estimates.

Factorial Structure of the Measurement Model													
Model No.	Model	χ^2^	df	CFI	TLI	RMSEA	SRMR	Model Comparison No.	Δχ^2^	Δdf	*p*	ΔCFI	ΔRMSEA
M1	2-factor model	1994.632	213	0.961	0.949	0.048	0.051						
M2	1-factor model	9382.963	225	0.799	0.753	0.105	0.108	M1	7388.331	12	<0.001	0.162	0.196
M3	4-factor model	Non-converged											
Longitudinal Measurement Invariance of the Hypothesised 33-factor Model													
M4	Metric Invariance	2014.547	225	0.961	0.952	0.046	0.051	M1	19.915	6	0.0687	0	0.003
M5	Strong Invariance	2130.083	241	0.958	0.952	0.046	0.053	M4	115.536	16	<0.001	−0.003	0
M6	Strict Invariance	2259.832	257	0.956	0.953	0.046	0.053	M5	129.749	16	<0.001	0.002	0

Note: N = 3694; χ^2^ = chi-square; df = degrees of freedom; CFI = comparative fit index; TLI = Tucker–Lewis’s index; RMSEA = root mean squared error of approximation; SRMR = standardised root mean squared residual.

**Table 3 ijerph-20-05280-t003:** Time invariance of the structural paths.

Analysis of the Alternative Structural Models													
Model No.	Model	χ^2^	df	CFI	TLI	RMSEA	SRMR	Model Comparison No.	Δχ^2^	Δdf	*p*	ΔCFI	ΔRMSEA
M7	Hypothesised RI-CLPM (unconstrained)	2218.475	254	0.957	0.953	0.046	0.052						
M8	M7 + autoregressive paths constraint equal across time	2219.153	256	0.957	0.953	0.046	0.052	M7	0.678	2	0.713	0.000	0.000
M9	M8 + paths from quantitative JI to qualitative JI constrained across time	2220.55	257	0.957	0.954	0.045	0.052	M8	1.400	1	0.237	0.000	0.001
M10	M9 + paths from qualitative JI to quantitative JI constrained across time	2221.26	258	0.957	0.954	0.045	0.052	M9	0.702	1	0.402	0.000	0.000

Note: N = 3694; χ^2^ = chi-square; df = degrees of freedom; CFI = comparative fit index; TLI = Tucker–Lewis’s index; RMSEA = root mean squared error of approximation; SRMR = standardized root mean squared residuals.

## Data Availability

Restrictions apply to the availability of these data. Data was obtained from Jill Nelissen from KU Leuven and are available upon request.

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
