# Peer review of "A Person-Centered Approach to Job Insecurity: Is There a Reciprocal Relationship between the Quantitative and Qualitative Dimensions of Job Insecurity?"

_ijerph, 2023, doi:10.3390/ijerph20075280_

Round 1
Reviewer 1 Report
The study proposes and tests the reciprocal relationship between quantitative and qualitative job insecurity at the within-person level. Authors used appropriate research methods and rely on a wide range of relevant literary sources. They tested three plausible types of association derived from the two most dominant theories used in the field of job insecurity. This study brings new knowledge and contributes to the ongoing discussion on how job insecurity dimensions influence each other. Based on the study research can continue the further research in this area.
Author Response
Dear Reviewers,
We sincerely thank you all for the time and effort you put into reviewing our article. Your observations and comments have confronted us with highly relevant issues that were unintentionally overlooked. We are confident that your feedback has helped us to improve the current study. We attach the comments we received from all three reviewers. To each comment, we provide a concise response, and we point to the section of the article that includes the alterations. We explain the context and the reasoning behind every questioned decision, as well as the changes in the text we made, following your recommendations. We hope you find these improvements to upgrade the manuscript sufficiently and make it ready for publication.
Best regards,
The Authors

Reviewer 2 Report
As you write in the introduction, job insecurity is a growing problem because the conditions for work and organizations are constantly changing. I was therefore surprised that the result and your conclusions do not link back to these global conditions. Instead, your conclusion is that the research should be directed towards person-centered theories about job insecurity. It's hard for me to understand how this is supported in the text.
Unfortunately, there are big issues with the content, both structurally and in terms of presentation, and it is difficult to follow the consistent theme. To begin with, it is unclear what is intended to be the knowledge gap to be investigated here. Initially, it is mentioned that the labor market with constant changes in organizations drives job insecurity. However, this soon disappears from view in the text and questions arise about what is the real problem that is meant. In other words, the formulation of the problem that should lead to the purpose and questions is not clearly formulated.
The concept of job insecurity are defined relatively late. On the other hand, a theoretical reasoning concerning how the theory intended to be employed occurs too early in the text. It is unclear whether the article is empirically driven or theory driven. 3 theories seem to be used here which between themselves have large discrepancies. For example deprivation theory focus on contextual factors and person-centered theory focus on individual factors to explain job insecurity. The logic in these choices is missing and early in the introduction it is emphasized that these theories intend to be used. However, the question quickly arises whether there is a purpose to test these theories? The text is also overloaded with different concepts and spreads without there being any connection between the different theoretical reasonings, which between them have completely different basic assumptions. If so, there is a problem with the design. Other variables or factors should be accounted for such as organizational change. 2 of the hypotheses in the study are linked to different theories, but it is then unclear how these theories can be linked to the result. What has been tested is the connection between 2 variables: qualitative job insecurity and quantitative job insecurity and how these relate to each other. The difference between these variables is described as uncertainty regarding a change in work tasks (qual) and uncertainty about losing one's employment (quant). 2 different estimation scales have been used, one for each variable. Of course, one would expect a review of the theories, but it is done with no more than a few words. There is no validity discussion about the variables and it is a bit difficult to see how these two types of uncertainty are important to compare. Is it possibly different aspects of the same phenomenon? It risks a tautological reasoning (that is job insecurity leads to job insecurity) and it can to some extent explain the non-relationship between the two investigated variables, as is pointed out in the text: "Bringing these results together, we found no evidence for an intraindividual cycle between quantitative and qualitative job insecurity."
If the aim was not intended to test theories, it can be assumed that the theories were chosen with the aim of explaining the result. Why then 3 different theories that is not compatible?
Measurement took place 3 times over an 18-month period on employees within 14 organizations. This is a bit hard to view as a longitudinell study as is claimed. Organizational change goes on over long periods of time and may continue to stir insecurity. This is not accounted for and you also claim to find causal relationships, which should be avoided in general with regard to behavioral science studies. It is above all, correlations between different variables or factors that can be the basis for generalization in behavioral science, not causality. Especially not when it comes to vaguely defined concepts, which is emphasized to be the case with job insecurity.
The result appears to be a rejection of previous theory (deprivation theory?), but it is not clearly stated how you came to that conclusion from the findings. Above all, a common thread is needed that makes the reasoning logical and comprehensible to the reader through all parts of the text.
Author Response
Dear Reviewer,
Thank you for your constructive feedback, which has helped us significantly improve the manuscript. In the attachment, we respond to each of your comments and point to the section of the article where we implemented changes. Considering the English language and style, they were revised by a professional proofreading editor.
Best regards,
The Authors

Reviewer 3 Report
This paper uses a rather new statistical approach (random-intercept cross-lagged panel model, RI-CLPM) to analyze the within-person relationship between quantitative and qualitative job insecurity (quant and qual JI) over time. I enjoyed reading it and I am sure others in the community will as well, and I have only some smaller comments to offer.
1) It might be an unusual comment, but I think there were too many theoretical arguments and not enough psychology. I agree with how the authors described the theories (i.e., Jahoda’s and Hobfoll’s) and their implications, but I am missing a more psychological story that explains what happens. Despite reading the paper twice, I find it difficult to imagine the psychological processes. Let’s assume there is a participant who works in one of these companies surveyed by the authors. He fears that there might be a restructuring coming which would mean that he loses his job or at least job features – that is the situation at t1. Now let’s move on in time - it is now 6 months later (by the way, why 6?). And now what? Because he considered his job in danger 6 months ago, does he now fear (because of high quant JI 6 months ago) that his job features are now particularly threatened? While still assuming that his job is danger? Does this mean all the worries 6 months ago make him generally more worrying? Or have the restructuring rumors increased and that’s why he experiences more quant and qual JI? I am not sure. And I would really appreciate if the authors explain in much more details what is potentially happening in participants’ head.
2) I think it is crucial to stress that the means of quant JI and qual JI stay the same across the three points in time because it is plausible that that variables correlate although the means change. For example, the relative standing of participants in quant JI might be similar over time but there might have been an overall decrease in quant JI over time. This is not the case here, and that’s good because how could you explain an overall increase and within-person correlations at the same? At the same time, I wonder whether the SDs should not increase over time: If quant JI correlates over time and within-person, shouldn’t be the distribution of quant JI become more extreme because those with high quant JI become even more concerned about their quant JI and those with low quant JI become even less concerned?
3) Do the authors have any information who really lost their job? I saw the drop-out analysis, but they do not say anything about jobs being lost. Furthermore, do the authors know in what financial situation the 13 organizations were – did they announce any job losses? These pieces of information seem crucial to understand changes in quant and qual JI over time.
4) On lines 396 ff., the authors describe how much variance is explained on the higher (between-person) and the lower level, but they do not mention that the lower level also includes the error variance. Thus, if they find 37% of the variance of quant JI on the lower level, this does not mean that this figure of 37% only captures within-person variance. Actually, it is often mainly error variance.
Minor issues
5) p. 6: please explain the abbreviations BQNi and WFQNi.
6) Table 1: please explain all variables (and their scoring) in the Notes. And given the N, I’d rather ignore p<.05 and p<.01.
7) Figure 1: please include the weights from latent factors to manifest variables.
8) Be kind to readers who skip over results section and just read intro and discussion – in other words, please give a summary of your results in your first paragraph of your discussion (i.e., instead of explaining (and repeating) what you expected, tell the reader what you found).
I hope the authors find my feedback helpful.
Author Response
Dear Reviewer,
Thank you for your kind words. We are happy that you enjoyed the article and recognize the added value to the field of job insecurity. We appreciate your constructive feedback that helped us to improve the manuscript. We respond separately to each comment below.
Best regards,
The Authors
1) It might be an unusual comment, but I think there were too many theoretical arguments and not enough psychology. I agree with how the authors described the theories (i.e., Jahoda’s and Hobfoll’s) and their implications, but I am missing a more psychological story that explains what happens. Despite reading the paper twice, I find it difficult to imagine the psychological processes. Let’s assume there is a participant who works in one of these companies surveyed by the authors. He fears that there might be a restructuring coming which would mean that he loses his job or at least job features – that is the situation at t1. Now let’s move on in time - it is now 6 months later (by the way, why 6?). And now what? Because he considered his job in danger 6 months ago, does he now fear (because of high quant JI 6 months ago) that his job features are now particularly threatened? While still assuming that his job is danger? Does this mean all the worries 6 months ago make him generally more worrying? Or have the restructuring rumors increased and that’s why he experiences more quant and qual JI? I am not sure. And I would really appreciate if the authors explain in much more details what is potentially happening in participants’ head.
Response: Thank you for your insightful observation. We now include an example of what might happen in employees’ heads when they are facing a threat to job loss or a threat to job characteristics. However, we would like to emphasize that the process of experiencing job insecurity and the relationship between the two aspects is complex and probably hard to envision. Thus, our example is a simplified version of what might be the actual process. We hope that the example improves the clarity of our objectives and helps to envision the psychological processes we test.
Changes to the article: Introduction. Literature review
2) I think it is crucial to stress that the means of quant JI and qual JI stay the same across the three points in time because it is plausible that that variables correlate although the means change. For example, the relative standing of participants in quant JI might be similar over time but there might have been an overall decrease in quant JI over time. This is not the case here, and that’s good because how could you explain an overall increase and within-person correlations at the same? At the same time, I wonder whether the SDs should not increase over time: If quant JI correlates over time and within-person, shouldn’t be the distribution of quant JI become more extreme because those with high quant JI become even more concerned about their quant JI and those with low quant JI become even less concerned?
Response: Thank you for this comment. Indeed, it is important to stress that the means of quantitative and qualitative job insecurity stayed nearly the same across the three time points, which suggests that employees in our sample on average experience no change in job insecurity. However, when we split the variance for the between-person part and within-person part, we observe that around 30% of the variance is at the individual level. This means that employees do experience change in felt job insecurity, overtime. At the same time the correlations between quantitative and qualitative job insecurity at those two levels are independent and informative in two-way: the correlation at the between person level tells us that at the population level employees who usually feel more threat to job loss also perceive a higher threat to job characteristics. The correlation at the within-person level informs us that employees who experienced a higher-than-usual threat to job loss are likely to simultaneously experience a higher-than-usual threat to their job characteristics.
I think that in the hypothesized case, when the means of quantitative and qualitative job insecurity would change across the three time points (decrease or increase), it is still possible (or even expected) to observe an overall increase or decrease in the means of quantitative and qualitative job insecurity, as well as simultaneously significant within-person correlations. Specifically, in the article we argue that the reason for the overall increase or decrease might be mostly due to time-varying predictors, thus at the within-person level (f.i. the sample would consist of employees whose organization goes through a merge or downsizing, and they all experience higher-than-usual job insecurity). Thus, whereas the relative standing of participants is similar overtime (between-person variance), at the same time they all experience a higher-than-usual level of job insecurity, which leads to higher means of job insecurity at the later time points.
Changes to the article: Results. We clarify that the means od quantitative and qualitative job insecurity stay the same across the three points and what it means.
3) Do the authors have any information about who really lost their job? I saw the drop-out analysis, but they do not say anything about jobs being lost. Furthermore, do the authors know in what financial situation the 13 organizations were – did they announce any job losses? These pieces of information seem crucial to understand changes in quant and qual JI over time.
Response: Unfortunately, we lack information about whether our participants lost their job during the observation period. Furthermore, we have no information on the financial situation of the companies that participated in the study. From the dropout analysis we know that respondents who dropped out did not significantly vary from the respondents who stayed throughout the observation period. Unfortunately, it is not possible to know why employees dropped out of the study and if it was due to the job loss. However, we asked our participants whether they have experienced a change to the job 6 months prior to the study. Over 60% of respondents did not expect any changes, which might suggest that the organizations that participated in our study were financially stable. Furthermore, the study was conducted in Belgium which has modern, capitalist and stable economy with a low unemployment rate. The majority of respondents worked full-time on a permanent contract. Overall, we reckon that respondents experienced relatively stable job conditions.
Changes to the article: Measurement. We add information about the changes at work that employees experienced prior to the study.
4) On lines 396 ff., the authors describe how much variance is explained on the higher (between-person) and the lower level, but they do not mention that the lower level also includes the error variance. Thus, if they find 37% of the variance of quant JI on the lower level, this does not mean that this figure of 37% only captures within-person variance. Actually, it is often mainly error variance.
Response: Thank you for that comment. We agree that the former method used did not account for measurement error. We did a literature search to find a way to account for this, and we found an article that introduces the reliability-adjusted intra-class correlation coefficient ICC(1). We implemented this method in our study and rewrote the section of the analysis. Here is the article we refer to: doi:10.3389/fpsyg.2020.00825; doi:10.1007/s11031-021-09867-5.
Changes to the article: Analysis. Results
Minor issues
5) p. 6: please explain the abbreviations BQNi and WFQNi.
Changes to the article: “Specifically, the model splits the variance of each variable into a time-invariant, trait-like part (the individual’s average level of the experienced quantitative and qualitative job in-security; between-person variance for quantitative and qualitative job insecurity─ BQNi and BQLi, respectively ) and a time-varying, state-like part that captures dynamic, overtime fluctuations around the individual’s expected score (within-person variance for quantitative and qualitative job insecurity─ WQNit and WQLit). The latter is used to estimate the autoregressive and cross-lagged effects, which test the hypotheses formulated above.” In summary, B refers to ‘between-person’ and W to ‘within-person’ variance. Thank you for pointing this out, which helped us to clarify this for the readership.
6) Table 1: please explain all variables (and their scoring) in the Notes. And given the N, I’d rather ignore p<.05 and p<.01.
Changes to the article: Table 1
7) Figure 1: please include the weights from latent factors to manifest variables.
Changes to the article: Figure 1
8) Be kind to readers who skip over results section and just read intro and discussion – in other words, please give a summary of your results in your first paragraph of your discussion (i.e., instead of explaining (and repeating) what you expected, tell the reader what you found).
Changes to the article: “The objective of this project was to identify the relationship between quantitative and qualitative job insecurity. We implement conservation of resources (COR) theory to propose a reciprocal relationship. Furthermore, we conduct a stepwise procedure to separately examine each direction in which quantitative and qualitative job insecurity might be related to one another. To answer our research question, we performed a multiple indicator random-intercept cross-lagged panel model (RI-CLPM). We used three-wave longitudinal data, collected every 6 months from the Belgian-employed population. The results are consistent with prior research, as we find a positive and significant correlation between quantitative and qualitative job insecurity, at the between-person level. This means that employees who on average report a higher level of quantitative job insecurity are more likely on average to report a higher level of qualitative job insecurity. In addition, quantitative and qualitative job insecurity are found to be significantly related at the within-person level, which indicates a positive interaction between an experienced change in quantitative and qualitative job insecurity. In other words, an employee who at ty experienced a higher-than-usual (or a lower-than-usual) threat to job loss was more likely to simultaneously experience a higher-than-usual (or a lower-than-usual) threat to job features. Lastly, the cross-lagged analysis of the relationship between quantitative and qualitative job insecurity failed to confirm a reciprocal relationship. The results indicate a unidirectional relationship, showing a significant longitudinal association between quantitative job insecurity and qualitative job insecurity. A higher-than-usual threat to job loss was positively related to a higher-than-usual threat to job characteristics, six months later.”
I hope the authors find my feedback helpful.

Round 2
Reviewer 2 Report
- The headline of the paper still indicate that the articles intension is suggestive towards a person-centered approach for job insecurity.
- Earlier research of work environment impact on job insecurity is neglected. Discussing perspectives on research needs a thorough reflecting on the perspective rejected.
- Also there is not a clear logic why this factors or scales has been chosen for the study.
- The findings doesn't add any solid contribution to the understanding of job insecurity.
Author Response
Dear Reviewer,
I would like to thank you all for the time and effort you put into reviewing our article. We hope that with the revisions, which we have made upon the request from the Academic Editor, you will find the manuscript more comprehensible.
Best regards,
The Authors